# An image-based flow cytometric approach to the assessment of the nucleus-to-cytoplasm ratio

Joseph A. Sebastian[1,2,3]*, Michael J. Moore[1,2,3], Elizabeth S. L. Berndl[1,2,3], Michael C. Kolios[1,2,3]*

1 Department of Physics, Ryerson University, Toronto, Canada, 2 Institute of Biomedical Engineering, Science and Technology (iBEST), A Partnership Between Ryerson University and St. Michael's Hospital, Toronto, Canada, 3 Keenan Research Centre for Biomedical Science, Li Ka Shing Knowledge Institute, St. Michael's Hospital, Toronto, Canada

* j.sebastian@mail.utoronto.ca (JAS); mkolios@ryerson.ca (MCK)

## Abstract

The nucleus-to-cytoplasm ratio (N:C) can be used as one metric in histology for grading certain types of tumor malignancy. Current N:C assessment techniques are time-consuming and low throughput. Thus, in high-throughput clinical contexts, there is a need for a technique that can assess cell malignancy rapidly. In this study, we assess the N:C ratio of four different malignant cell lines (OCI-AML-5—blood cancer, CAKI-2—kidney cancer, HT-29—colon cancer, SK-BR-3—breast cancer) and a non-malignant cell line (MCF-10A –breast epithelium) using an imaging flow cytometer (IFC). Cells were stained with the DRAQ-5 nuclear dye to stain the cell nucleus. An Amnis ImageStreamX® IFC acquired brightfield/fluorescence images of cells and their nuclei, respectively. Masking and gating techniques were used to obtain the cell and nucleus diameters for 5284 OCI-AML-5 cells, 1096 CAKI-2 cells, 6302 HT-29 cells, 3159 SK-BR-3 cells, and 1109 MCF-10A cells. The N:C ratio was calculated as the ratio of the nucleus diameter to the total cell diameter. The average cell and nucleus diameters from IFC were 12.3 ± 1.2 µm and 9.0 ± 1.1 µm for OCI-AML5 cells, 24.5 ± 2.6 µm and 15.6 ± 2.1 µm for CAKI-2 cells, 16.2 ± 1.8 µm and 11.2 ± 1.3 µm for HT-29 cells, 18.0 ± 3.7 µm and 12.5 ± 2.1 µm for SK-BR-3 cells, and 19.4 ± 2.2 µm and 10.1 ± 1.8 µm for MCF-10A cells. Here we show a general N:C ratio of ~0.6–0.7 across varying malignant cell lines and a N:C ratio of ~0.5 for a non-malignant cell line. This study demonstrates the use of IFC to assess the N:C ratio of cancerous and non-cancerous cells, and the promise of its use in clinically relevant high-throughput detection scenarios to supplement current workflows used for cancer cell grading.

## Introduction

There is a need in cancer diagnostics for techniques that overcome the drawbacks of current conventional cancer cell assessment methods. Currently, histological assessment is the gold standard of assessing cell and tissue malignancy [1] but lacks speed, high-throughput, and can

**Data Availability Statement:** All data files are available on Zenodo (DOI: 10.5281/zenodo.4552851).

**Funding:** This research was supported in part by the Natural Science and Engineering Research

Council of Canada (https://www.nserc-crsng.gc.ca/index_eng.asp) Discovery Grant (RGPIN-2017-06486), the Canadian Foundation for Innovation (https://www.innovation.ca/) and the Ontario Ministry for Research and Innovation (Project #11525), and the Terry Fox Foundation (https://terryfox.org/, TFRI Project #1034) funding agencies. All grants were awarded to M.C.K. The funders had no role in study design, data collection and analysis, decision to publish, or preparation of the manuscript.

**Competing interests:** M.C.K. and M.J.M. have financial interests in Echofos Medical Inc., which did not support this work. The remaining authors declare no competing financial interests. This does not alter our adherence to PLOS ONE policies on sharing data and materials.

be prone to differing interpretation from pathologists. In addition, histology is an inefficient technique in clinically relevant contexts that require high-throughput cellular analysis such as diagnostics of hematological diseases [2], diagnosis of minimal residual disease [3] and circulating tumor cell detection [4]. Thus, the utilization of high throughput techniques to assess cell malignancy may improve cancer diagnostics by providing cellular information of these rare phenotypes.

Clinicians have identified an enlarged nucleus as a prevalent characteristic of certain types of malignant cells [5–7]. The enlarged nucleus of these malignant cells led to the development of the nucleus-to-cytoplasmic (N:C) ratio, defined as the ratio of the cross-sectional area of the nucleus divided by that of the cytoplasm [8]. Although histology and cytology are the gold standards of the N:C ratio assessment method, many studies have reported the interobserver variability that exists in during visual quantitation [9–11]. In practice, cytology, where slides of biological specimen are fixed to a glass slides and examined, is another method used to assess the N:C ratio. In addition, although this metric is used in many tissue types (e.g., urothelial carcinoma), in others (e.g., melanoma), lower N:C ratios [12] despite malignancy and higher N:C ratios in normal cells (e.g., lymphocytes) prevent adoption of the N:C ratio as a grading method. Nevertheless, new techniques with less subjectivity have been developed to assess and quantify the N:C ratio of cancer cells (e.g., computer vision, two-photon microscopy, immunohistochemistry analysis techniques) [13–18]. Since these techniques rely on histological sectioning to assess the N:C ratio, they lack translatability to high-throughput clinical contexts where a liquid biopsy would be used to assess malignancy. A high-throughput technique that can assess cell malignancy using cells in suspension would be ideal. Imaging flow cytometry (IFC) can provide such a cytometric assessment method due to the high-throughput collection of images of single cells. IFC is a hybrid technology that combines conventional flow cytometry (FC) with high-throughput microscopy to generate high-resolution images of single cells in suspension within minutes [19, 20]. IFC combines the advantages of using FC with the ability to visually identify single cells. IFC has been used in many applications, including radiation biodosimetry, analysis of autophagy, and quantification of cellular heterogeneity [21–24]. In this work we demonstrate how IFC can be used to assess the N:C ratio of several malignant cell lines and a single non-malignant cell line.

Previously, our group has used IFC to compare cell size measurements with measurements done by photoacoustic microscopy and photoacoustic flow cytometry which were used for the N:C analysis of cultured breast and prostate cancer cells [25–27]. In this work, we determine the N:C ratio of four different malignant cell lines each originating from *different* tissues and the N:C ratio of a single non-malignant cell line. Here, acute myeloid leukemia (OCI-AML-5, blood cancer), CAKI-2 (kidney cancer), HT-29 (colon cancer), SK-BR-3 (breast cancer) cells, and MCF-10A (breast epithelial) cells were used for the measurements. Across cell lines, we observe varying cell and nuclear sizes but a common N:C ratio of ~0.6–0.7, consistent with international standards of diagnosis of urothelial carcinoma [7] and a smaller N:C ratio of 0.53 in the non-malignant cell type. This work demonstrates the diagnostic potential of IFC as an assessment technique of the N:C ratio and the promise of its use in clinically relevant high-throughput detection scenarios.

## Methods

### Cell preparation

In this work (1) SK-BR-3 (ATCC, Virginia, USA, HTB-30), HT-29 (ATCC, Virginia, USA), and CAKI-2 cells were thawed and cultured for two weeks in McCoy's 5A (modified) Media (Wisent Inc., Quebec, Canada, 317-010-CL) supplemented with 10% fetal bovine serum (FBS),

and 1% penicillin/streptomycin, (2) OCI-AML-5 cells (DSMZ, Braunschweig, Germany, ACC247) were thawed and cultured for two weeks in Alpha modified Eagle medium (Wisent Inc., Quebec, Canada, 310-010-CL) containing 10% FBS and 1% penicillin-streptomycin (Wisent Inc., Quebec, Canada, 450-201-EL) by volume, and (3) MCF-10A cells MCF-10A cells (Addex Bio, California, USA, C0006015) were thawed and cultured for three weeks in 1:1 Dulbecco's Modified Essential Media and F12 media (ThermoFisher, Massachusetts, USA, 11330032), supplemented with 5% Horse serum (Sigma-Aldrich, Ontario, Canada, H1270), 20 ng/mL EGF (Peprotech, New Jersey, USA, AF-100-15), 0.5 ug/mL Hydrocortisone (Sigma-Aldrich, Ontario, Canada, H0888), 100 ng/mL Cholera Toxin (List Biological Laboratories, California, USA, 100B), 10ug/mL Insulin (Sigma-Aldrich, Ontario, Canada, I1882), and 1% penicillin-streptomycin (Wisent Inc., Quebec, Canada, 450-201-EL). Once confluent, all adherent cells were trypsinized and resuspended in phosphate-buffered saline (PBS). The cells were stained using a 1:200 solution of DRAQ-5 (Thermo Fisher, Mississauga, Canada), a fluorescent nuclear dye and resuspended in a volume of 50 μl PBS and 1% FBS in a 1.5 ml low retention microfuge tube (MilliporeSigma, Oakville, Canada).

## Image flow cytometer operating parameters and post-processing

The imaging parameters and post-processing for this work have been previously described [26]. Briefly, An Amnis ImageStreamX® MarkII IFC (MilliporeSigma, Seattle, USA) equipped with a 5-laser 12-channel system at 60x magnification, following ASSIST calibration (MilliporeSigma, Seattle, USA) was used for image acquisition. In this study, channels 1 to 11 were used for acquisition along with a 642-nm laser (150 mW); however, analysis on only channel 1 (430 to 480 nm), 7 (430–505 nm), and 11 (660–740 nm) were completed (Ch 1/7 – malignant cells, Ch1/11 –non-malignant cells). Since both Channel 7 and 11 are used for fluorescent imaging in IFC, there is no difference between usage of either channel. A bright-field area lower limit of 50 μm$^2$ was used to eliminate debris and speed beads during acquisition.

Cell image analysis was carried out using the Amnis IDEAS® software platform (version 6.2). An overview of the analysis workflow is shown in Fig 1A. The nucleus diameter and cell diameter were determined using a custom workflow in IDEAS, which is illustrated in Fig 1 and adapted from our previous work [26]. Briefly, from Fig 1.a.I, the gradient root-mean squared feature was applied to the acquired OCI-AML-5, CAKI-2, HT-29, SK-BR-3, and MCF-10A cell images, and the corresponding values were plotted on a normalized relative frequency distribution to remove unfocused cell images. Fig 1.a.II depicts the area and aspect ratio features combined to remove images containing multiple cells. In our workflow, we included cell images with an aspect ratio between approximately 0.55 and 1 to avoid cell fragments and other debris. Fig 1.a.III shows the raw centroid X feature plotted against a normalized relative frequency distribution to remove clipped cell images. The raw centroid X feature quantifies the central location of the acquired images. Lastly, Fig 1.a.IV depicts a positive gate for DRAQ-5-positive cells that was obtained using fluorescence intensity and area features. Through gating for solely DRAQ-5-positive cells in plot IV, we exclude cell images containing calibration beads, which are required for alignment of the sample stream during imaging. The rationale for this gate is based on the visual clustering of cells we see in the scatter plot based on area and high pixel intensity. Masks used for the image analysis process following the same protocol as our previously published work to accurately measure cell diameter and can be seen in Fig 2B–2D [25–27]. Here, the eroded masks were used to determine the cell diameter where IDEAS provided the diameter of a circle that has the same area as the eroded masks [28]. DRAQ-5 leakage out of the nucleus was addressed in our workflow for the non-malignant cells by adding a plot that used gradient root-mean-squared feature on the fluorescent images to isolate cells that had

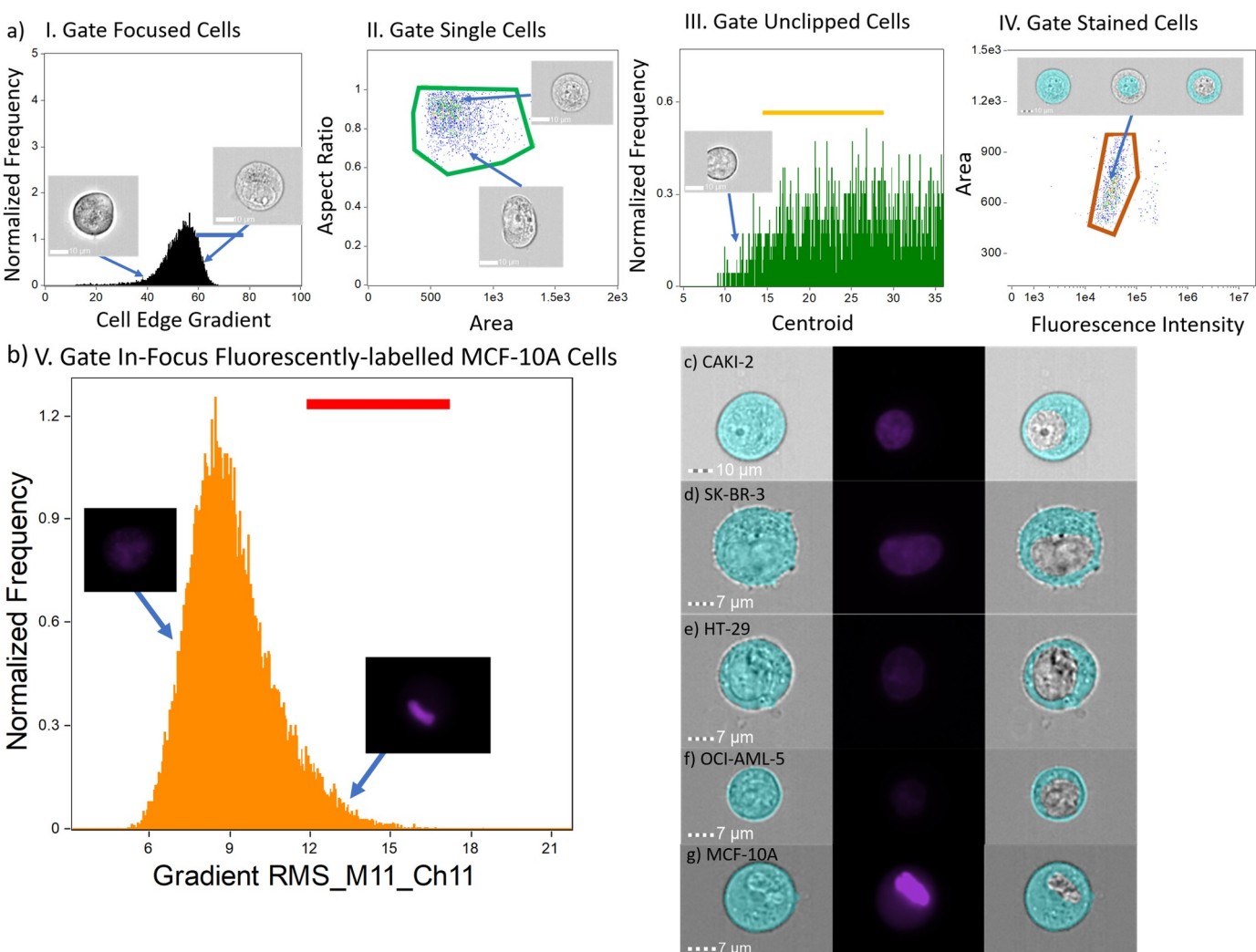

**Fig 1. IFC analysis workflow and representative cells.** (a) The IFC analysis workflow excludes: (I) unfocused cells, (II) multiple cells per brightfield image, (III), clipped cells in brightfield images, (IV) unstained cells. Sequential gating bars (I–blue, II–green, III–yellow, IV–orange) depict manually gated ROIs for each cell population. (b) Additional gating step for MCF-10A cells to address leakage due to extended trypsinization. (c-g) Representative cells from each cell line after IFC gating cell mask (left column), stained fluorescent image (middle column), and cytoplasm (right column).

localization of the DRAQ-5 dye in their nucleus (high gradient RMS value). This plot is provided in Fig 1B. We applied this extra gate on all cell lines and saw minimal changes in the N:C ratio across the cancer cell lines (see S1 Table). The feature statistics from each measurement of each cell line were exported from IDEAS and imported into Jupyter Notebooks. Further postprocessing involved the removal of images of cells which were analyzed as having nucleus diameters larger than that of the entire cell diameter. Lastly, the N:C ratio for the leftover cells was calculated. All data and post processing scripts can be found here.

## Results

### Image flow cytometer

In this work, we interrogated four diverse cancer cell lines: OCI-AML-5, a blood cancer cell line; CAKI-2, a kidney cancer cell line, HT-29, a colon cancer cell line; and SK-BR-3, a breast

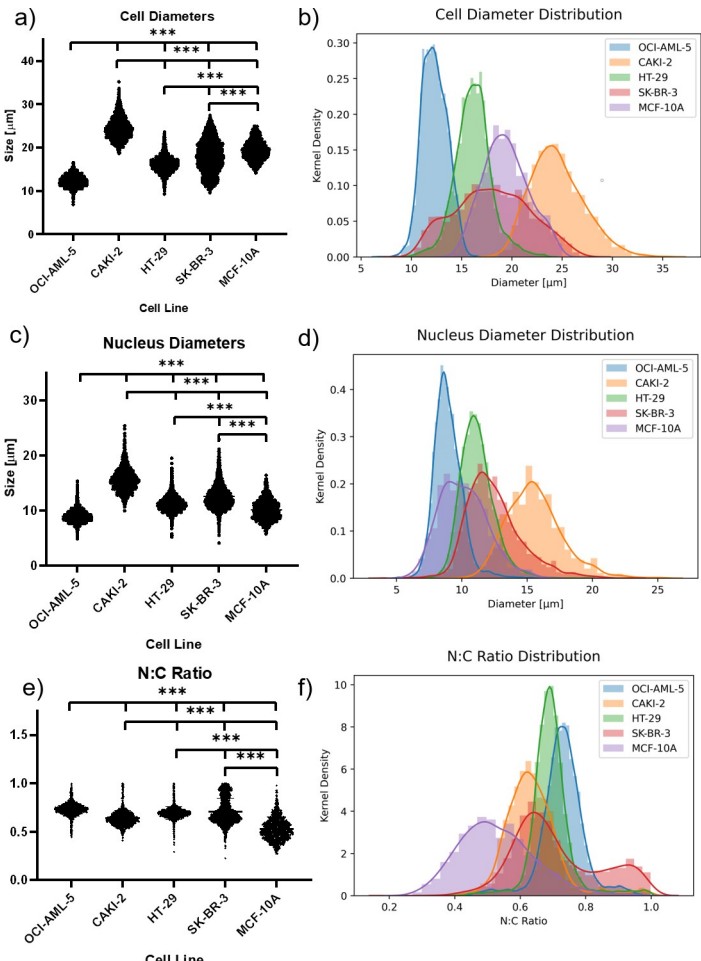

**Fig 2. Cell diameter, nuclear diameter, and N:C ratio distribution.** (a, c, e) Beeswarm plots and (b, d, f) density distributions display (a, b) cell, (c, d) nucleus, and (e, f) N:C ratio distribution across cell lines. Triple asterisks indicate a *p*-value of <0.001.

cancer cell line. Initially, 43,237 OCI-AML-5, 19,699 CAKI-2, 32,907 HT-29, and 25,073 SK-BR-3 cells were imaged using the IFC within minutes. In our IFC gating process, we excluded out of focus cells, cell images containing alignment beads, and multiple or fragmented cells. We also removed objects analyzed in the IDEAS software that indicated a larger nucleus mask than cytoplasm mask. From this post-processing, we were able to analyze 5284 OCI-AML-5 cells, 1096 CAKI-2 cells, 6302 HT-29 cells, and 3159 SK-BR-3 cells. Table 1 and Fig 1B–1F provides a summary of the results of the IFC gating, analysis and post processing for each cell line and corresponding standard deviations. There is noticeable variability in the nuclear uptake of the DRAQ-5 as depicted in the 2nd vertical panel of Fig 1C–1G. The

**Table 1. A summary of IFC measurements and corresponding standard deviations for the cell and nucleus diameter as well as the N:C ratio for each cell line.**

| Cell Line | OCI-AML-5 (n = 5284) | CAKI-2 (n = 1096) | HT-29 (n = 6302) | SK-BR-3 (n = 3159) | MCF-10A (n = 1109) |
|---|---|---|---|---|---|
| **Cell Diameter [μm]** | 12.3 ± 1.2 | 24.5 ± 2.6 | 16.2 ± 1.8 | 18.0 ± 3.7 | 19.4 ± 2.2 |
| **Nucleus Diameter [μm]** | 9.0 ± 1.1 | 15.6 ± 2.1 | 11.2 ± 1.3 | 12.5 ± 2.1 | 10.1 ± 1.8 |
| **N:C** | 0.73 ± 0.07 | 0.64 ± 0.08 | 0.69 ± 0.07 | 0.71 ± 0.13 | 0.53 ± 0.11 |

variability of nuclear dyes has been studied and empirically determined as being cell cycle, cell-type, and concentration dependent [29–31]. To account for this variability in dye uptake, the fluorescent intensity gating process (Fig 1.a.IV) is individualized for each cell line. We also interrogated 158,069 MCF-10A non-malignant breast epithelial cells. Following the post-processing steps outlined above on the malignant cells, we were left with 21,396 cells that were DRAQ-5 positive but overestimated the N:C ratio of these cells. However, in looking at the IFC images, we noticed significant leakage of the nuclear dye out of the nucleus into the cell. Thus, we gated the fluorescent channel images based on their gradient root-mean-squared values to isolate cell images that were highly focused to distinguish between the nucleus and other parts of the cell (see Fig 1B). This left us with 1,109 cells MCF-10A cells. This modified template produced minimal changes to the N:C ratio when applied on all four malignant cell lines (see S1 Table) but did lower the analyze sample population. Overall, CAKI-2 cells had the largest diameter (24.5 ± 2.6 μm), followed by MCF-10A cells (19.4 ± 2.2 μm), SK-BR-3 cells (18.0 ± 3.7 μm), HT-29 cells (16.2 ± 1.8 μm), and lastly, OCI-AML-5 cells (12.3 ± 1.2 μm). The IFC measurements of cell diameter are in good agreement with published values of the OCI-AML-5 and HT-29 cell diameters that used the Coulter Counter [32], slightly larger than SK-BR-3 cell sizing using microfluidic cytometry [33], and in good agreement with published values of MCF-10As [34]. Moreover, our previous studies with MCF-7, PC-3 and MDA-MB-231 cells were all in good agreement with other validated techniques [25–27]. The nuclear diameters of these cell lines follow a similar trend, with CAKI-2 cells (15.6 ± 2.1 μm) having the largest nuclei, followed by SK-BR-3 cells (12.5 ± 2.1 μm), HT-29 cells (11.2 ± 1.3 μm), MCF-10A cells (10.1 ± 1.8 μm), and OCI-AML-5 cells (9.0 ± 1.1 μm). From the cell and nuclear diameters, the N:C ratio of each cell line was calculated. All cancerous cell lines, regardless of tissue origin, had similar N:C ratio values. OCI-AML-5 cells had the largest N:C ratio (0.73 ± 0.07), followed by SK-BR-3 cells (0.71 ± 0.13), followed by HT-29 cells (0.69 ± 0.07), CAKI-2 cells (0.64 ± 0.08). The non-malignant MCF-10A cells had the smallest N:C ratio (0.53 ± 0.11). In addition, given the standard deviation of the MCF-10A cells, their N:C ratios are also within range of published values [35]. Corresponding beeswarm plots and kernel density histograms of the size distributions for the cell diameter (Fig 2A and 2B), nucleus diameter (Fig 2C and 2D), and N:C ratio (Fig 2E and 2F) are shown in Fig 2.

An ordinary one-way ANOVA with Tukey's multiple comparisons test (alpha = 0.05) was used to test for significant differences between the means of the cell, nuclear, and N:C ratios between cell lines (GraphPad Prism v8.0, San Diego, USA). A statistically significant difference ($p<0.001$) was observed between all means between cell lines.

## Discussion

The N:C ratio can be used as a histological metric in grading malignant disease in certain tissue types and cytologic specimens. In these biological samples, an enlarged nucleus has become a hallmark due to the abundance of chromatin present within malignant cells [1]. Currently, histology is the gold standard assessment method for the determination of the N:C ratio but it cannot be practically used when analyzing large populations of cells. Histology is advantageous when examining cohesive cells and tissue fragments, but it cannot be optimally used when analyzing large populations of cells. In addition, techniques which require a single cell suspension, such as IFC, would require special preparation techniques if using samples obtained from punch biopsies. These methods could in turn alter the cellular morphology and state of the cells and impact the resultant analysis. However, in many high-throughput clinical contexts, such as bodily fluid analysis for the detection of circulating tumor cells, minimal residual disease and hematological diseases, cytology would be cumbersome. Thus, an objective high-

throughput technique to assess the N:C ratio would provide an approach that would provide this measurement more reliably and with larger sample datasets.

The drawbacks of histology to assess the N:C ratio and reported inaccuracies and inconsistencies in N:C assessments by morphologists and clinicians [9–11] have motivated the implementation of new techniques to determine the N:C ratio, including the use of computer vision [13], multi-photon microscopy [14–16], Cell-CT [17] and immunohistochemistry analysis techniques [18]. Rahmadwati et al. [13] applied k-means clustering to segment nuclei and cytoplasm from background and connective tissue to detect features indicative of normal tissue, pre-cancerous tissue, and malignant tissues in cervical cancer histology images to assess the N:C ratio. Morphological features were extracted from a region-of-interest (ROI) on sample histological images of normal tissue, pre-cancerous tissue, and malignant tissues. These extracted features were used to classify other histology slides as normal, pre-cancerous, or malignant. The technique is heavily ROI dependent and details regarding sample size and number of histological images used are not provided. Multi-photon microscopy techniques studied by Huang et al. [14, 15] suffered from low sample populations (n < 25) and thus, were not representative of the entire cell population and ineffective in high-throughput settings. A two-photon microscopy (TPM) technique was implemented by Su Lim et al. [16] to assess the nuclear area and N:C ratios in human colon tissues. Although the authors analyzed thousands of TPM images from *ex vivo* colon histological slices for 7 patients, their technique would not be feasible in a high-throughput context. Moreover, the technique is specific to colon cancer. The Cell-CT [17] device has been used to assess the nucleus-to-cytoplasmic ratio but has a lengthy imaging time. This poses a problem for live cell imaging as the cells may undergo apoptosis or other alterations during imaging. Lastly, Xu et al. [18] used Image Pro Plus 6 (Media Cybernetics Corporation, USA) to calculate the nuclear/cytoplasmic ratio of 70 pairs of gastric cancer tissues, that are positive for death domain associated protein 6 (Daxx), and adjacent normal tissues. Three microscope images at 400x magnification were obtained for each tissue sample and each image included at least 100 Daxx positive cells. This clinical study shows the application of the current gold standard, histological sectioning, combined with a computational histological analysis method. The subjectivity and low-throughput nature of histology is improving using digital pathology and computer-aided image analysis; however, IFC provides an alternative for high throughput analysis with large sample analysis populations measured in a short period of time and would be useful in high-throughput clinical contexts to assess the N:C ratio of certain types of cells.

Image flow cytometry boasts high-throughput and multiparametric abilities enabling the acquisition of morphological information of single cells. The lack of high throughput techniques available to analyze the N:C ratio provides a niche opportunity for this emerging cytometric method to be used in certain cell types. Although IFC does take 2D images of single cells, the high-throughput nature of IFC provides a more reliable estimate of the N:C ratio over an analysis of a single 2D histological slice. Moreover, histological sectioning analysis requires microscope images from sections of a slice with an unknown amount of stained positive cells. By combining the statistical and gating capabilities of flow cytometry with the imaging capabilities of bright-field microscopy, IFC provides an opportunity for the development of a statistically powerful analysis (thousands of cells) of the N:C ratio in cancerous and nonmalignant cells. Here, we interrogated four cell lines of variable tissue origin (blood cancer, kidney cancer, colon cancer, and breast cancer) and a single non-malignant cell line (breast epithelium). Although the cell and nuclear diameters of each cancerous cell line differed (Fig 2) and statistical analysis revealed a significant difference between all their means, there is considerable overlap between their respective N:C ratios. However, we see a clear difference between malignant and non-malignant N:C ratios.

From Fig 2A and 2D, the measurements of cell diameter were consistent with the measurement of nuclear diameter over a range of cell sizes. This relationship between cell and nuclear size throughout the cell cycle is in line with what has been observed in the literature [36–38] and the future studies will examine the effect of the cell cycle [37] on the N:C ratio. In particular, the distribution of SK-BR-3 cell diameter found in Fig 2A and 2B does not show a Gaussian-like shape like the other cell lines did. The shape of the cell size distribution can be affected by the length of time the cell spends in each phase of the cell cycle. In this work however, the time in the S/G2/M phase is similar for all the cell lines studied (SKBR3–36%, HT-29–32%, CAKI-2–35%) [39–41]. Therefore, it is possible that the size variability of the SK-BR-3 cell line is inherent to this cell line.

To our knowledge, this high-throughput study is the first to measure the N:C ratio across malignant and non-malignant cells of different tissue origins. Our N:C results are consistent with international standards of cytopathology (Hang et al.'s [42] finding of an N:C ratio cutoff value of 0.5 for atypical urothelial cells, and McIntire et al.'s [43] finding of a N:C ratio cutoff value below 0.7 for high-grade urothelial carcinoma). Previous work by Rahmadwati et al. [13] and Huang et al. [14, 15] have shown that non-malignant cell types have N:C ratios between 0.2–0.4. Given the standard deviations of our non-malignant cell N:C ratios, our measurements fall within this range. To bolster our hypothesis related to the N:C distributions observed in our study, we plan on conducting a larger-scale study composed of multiple cell lines and their non-malignant counterparts. Additionally, since DRAQ-5 was the only nuclear dye examined in this study, the examination of the effects of various nuclear stains, their dye uptakes, and effects on the N:C ratio will be studied. However, we hypothesize that an alternate dye would not address the leakage problem as DRAQ5 has shown high nuclear localization in other studies [44]. Moreover, our future work will consider the effects of aneuploidy in the context of the N:C ratio. As many cell line subtypes can have varying nuclear size, shape, and complexity, their N:C ratio can be variable. It is beyond the capabilities of the currently presented technique to definitively state the cell subtype. However, a preliminary comparison between the N:C ratios of malignant and non-malignant cell lines does point towards a potential use for determining whether an individual cell is oncogenically transformed. Moreover, since this study focused on the presentation of a novel high-throughput technique to characterize the N:C ratio in multiple cell lines, the effects of aneuploidy were not considered.

IFC has some inherent disadvantages. Post-processing of the acquired cell images removed many images. Removal of images due to cells being out of focus, truncated cell boundaries, the presence of alignment beads in the field of view and all other factors resulted in the exclusion of up to 80–94% of acquired malignant cell images. Of the excluded images, 80–90% of the images were excluded during the gating step that determined whether the cell image was in focus. This exclusion was based on a cell edge gradient threshold of 60, commonly used in IFC gating analysis. The trypsinization of all cells before use in the IFC could have led to cytological damage to the cell membranes that caused their removal during the post-processing of the acquired IFC images In addition, we hypothesize that trypsin's known damaging effects on the cell membrane [45] may have led to cell fragmentation that could have led to exclusion of affected cells during the post-processing of the acquired IFC images and account for the high cell loss in our IFC post-processing analysis. For example, the post-processing steps that gate single cell IFC images could have removed cell images that contained both the cell and its corresponding fragment due to trypsinization Initially, using the malignant cell line workflow, we noticed DRAQ-5 leakage out of the nucleus in the analyzed IFC images. We hypothesize that the extended immersion in the trypsin solution could have caused damage to the cell membrane [45], proteomic alterations [46], decrease in cellular protein expression [47], and damage to the nuclear proteins [48]. To account for this leakage, we added an extra gradient-root-mean-squared gate to isolate fluorescent cell

images with sharp nuclear boundaries that were highly focused. This modified template produced minimal changes to the N:C ratio when applied on all four malignant cell lines (see S1 Table) but did lower the sample population that was analyzed. This subjective gating and masking approach is a limitation to imaging flow cytometry. Nevertheless, we are still able to reliably size thousands of cells. Modifications of this exclusion criteria could risk unfocused cells being used in proceeding gating steps and potentially impact the accuracy of our results. Moreover, cells with fragmented plasma membranes caused leaking of the DRAQ-5 into other parts of the cell which caused nuclear masks to be larger than cell masks. These cells were excluded in post-processing steps but demonstrates a limitation of the analysis technique to identify a stained nucleus if nuclear membrane fragmentation occurs. Similarly, the use of the "Diameter" feature to mask the cell or nucleus provides the diameter of a circle that has the same area as the masked cell or nucleus [28]. This step introduces the possibility of overestimating the nuclear diameter in cases of nuclear shape variability. Furthermore, although the gating strategies used in our analysis template were well-suited for the assessment of the N:C ratio, the potential user bias associated with the *development* of this template also introduces a subjective element in this approach. We have observed these common drawbacks to IFC in our previous work [49] and other groups have sought to improve the gating process [50, 51]. To eliminate these disadvantages of the IFC analysis workflow, our future work will focus on the use of computer vision and machine learning strategies to assess the N:C ratio. Here, no gating strategies are implemented to limit the loss of IFC images. Our group combined computer vision and machine learning strategies in the context of red blood cell storage lesions [52, 53]. We look to further incorporate techniques used by Doan et al. [54], Blasi et al. [55], and Hennig et al. [56] that use an unsupervised or weakly supervised, deep-learning method [57] for the assessment of the N:C ratio in both non-malignant and malignant cell lines. This would overcome drawbacks of IFC manual gating and user bias to provide an objective assessment of cell malignancy in a high-throughput context.

## Conclusion

We present a high-throughput image flow cytometric assay to assess the cell diameter, nuclear diameter, and N:C ratio of four *different* malignant cell lines (OCI-AML-5—blood cancer, CAKI-2—kidney cancer, HT-29—colon cancer, and SK-BR-3—breast cancer) and a single non-malignant cell line (MCF-10A –breast epithelium). We observe that, although the cells had a wide range of cell and nuclear sizes, a general N:C ratio of ~0.6–0.7 is common to all interrogated cancer cell lines and an N:C ratio of 0.53 for the non-malignant cell line. Limitations of our analysis technique lie in the manual gating strategies used in the accompanying analysis software. Our future work will focus on the application of computer vision and machine learning techniques on IFC data to assess this metric in rare cell types and more non-malignant cells.

## Supporting information

**S1 Table. A summary of IFC measurements and corresponding standard deviations for the cell and nucleus diameter as well as the N:C ratio for each cell line using the modified IFC workflow on all non-malignant and malignant cell lines.**
(DOCX)

## Acknowledgments

We gratefully acknowledge Michael Parsons for his key insights into the development of the gating template on the IDEAS software platform and his assistance with the Amnis Image-Stream Imaging Flow Cytometer.

## Author Contributions

**Conceptualization:** Joseph A. Sebastian, Michael J. Moore, Michael C. Kolios.

**Formal analysis:** Joseph A. Sebastian.

**Funding acquisition:** Michael C. Kolios.

**Investigation:** Joseph A. Sebastian, Elizabeth S. L. Berndl.

**Methodology:** Joseph A. Sebastian, Michael J. Moore, Elizabeth S. L. Berndl.

**Project administration:** Joseph A. Sebastian, Michael J. Moore, Elizabeth S. L. Berndl, Michael C. Kolios.

**Resources:** Elizabeth S. L. Berndl, Michael C. Kolios.

**Supervision:** Michael J. Moore, Michael C. Kolios.

**Visualization:** Joseph A. Sebastian.

**Writing – original draft:** Joseph A. Sebastian, Michael J. Moore, Elizabeth S. L. Berndl, Michael C. Kolios.

**Writing – review & editing:** Joseph A. Sebastian, Michael J. Moore, Elizabeth S. L. Berndl, Michael C. Kolios.

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
