## [Decision Letter · Decision Letter 0]

9 Nov 2020

PONE-D-20-26034

An image-based flow cytometric approach to the assessment of the nucleus-to-cytoplasm ratio

PLOS ONE

Dear Dr. Sebastian,

Thank you for submitting your manuscript to PLOS ONE. After careful consideration, we feel that it has merit but does not fully meet PLOS ONE’s publication criteria as it currently stands. Therefore, we invite you to submit a revised version of the manuscript that addresses the points raised during the review process.

As you can see from the reviews, one of the reviewers has very strong concerns about this manuscript.  Please address them before I can consider accepting  this manuscript. 

We look forward to receiving your revised manuscript.

Kind regards,

Jeffrey Chalmers, Ph.D.

Academic Editor

PLOS ONE

Journal Requirements:

"M.C.K. and M.J.M. have financial interests in Echofos Medical Inc., which did not support this work. The remaining authors declare no competing financial interests."

Reviewers' comments:

Reviewer's Responses to Questions

**Comments to the Author**

1. Is the manuscript technically sound, and do the data support the conclusions?

Reviewer #1: No

Reviewer #2: Yes

2. Has the statistical analysis been performed appropriately and rigorously? 

Reviewer #1: No

Reviewer #2: Yes

3. Have the authors made all data underlying the findings in their manuscript fully available?

Reviewer #1: Yes

Reviewer #2: Yes

4. Is the manuscript presented in an intelligible fashion and written in standard English?

Reviewer #1: Yes

Reviewer #2: Yes

5. Review Comments to the Author

Reviewer #1: The manuscript reports a method for assessing N:C ratios in 4 cancer cell lines using DRAQ5 as a nuclear stain and imaging flow cytometry. The cell lines are exemplars to illustrate the approach. In addition, the N:C ratio needs to be compared with normal cells. This is especially relevant if the intent is to use this strategy to detect CTC; in this situation data of N:C for normal nucleated cells in blood (i.e. leucocytes) should be assessed.

If this work is merely to describe the method, then the precise biology of each cell line is not relevant. This comment is made as the details of the cell lines are not provided. Are any of the cell lines aneuploidy? Of particular note is the “AML cell” line which has no name or details. There are many subtypes of acute myeloid leukaemia and they can be heterogeneous with varying nuclear size, shape and complexity by subjective appearance by light microscopy. These cells are likely to have a broad range of N:C ratios. The N:C ratio needs to be compared with normal cells. This is especially relevant if the intent is to use this strategy to detect CTC; in this situation data of N:C for normal nucleated cells in blood (i.e. leucocytes) should be assessed. Also SK-BR3 is a breast cancer cell line derived from the pleural fluid, a metastatic site. As such it may not be representative of the primary tumour but may be aneuploid.

The loss of >80% of cells in the analysis raises questions over the methodology, instrument set up, validity of the data and applicability in practice. Did the cell loss occur pre-analytical as a consequence of the trypsin causing cytological damage to cell membranes or is the loss due to instrument factors? Can the authors address this high cell loss? The cell edge gradient/focus and clipped cells data shows a high level of flow core instability (>60 cell edge gradient). This reviewer does not agree with the authors comment that this is a standard limitation or operation of imaging flow cytometry. This needs to be addressed.

DRAQ-5 stain as demonstrated in Figure 1 shows variable intensity and quality between the 4 cell lines. Were DRAQ-5 titrations performed on the cell lines to standardise masking and stain intensity measurements? How did the “masking” strategy take this variability into account? Is the variable intensity of DRAQ-5 stain related to the cell line, stage of cell cycle or ploidy? Were any other fluorescent nuclear stains assessed? Would alternate stain have addressed the leakage issue? Figure 2f shows SK-BR-3 cells with bimodal distribution of N:C ratios. Is this variability due to stain intensity, cell proliferation or cell viability? Have any other DNA stains been used? Would these have addressed the leakage issue? How has nuclear complexity been taken into consideration when the nuclear diameter was measured? The cell cycle and nuclear content was acknowledged as influencing the N:C results. How did different stages of cell cycle affect nuclear measurements and subsequent N:C ratio?

The nuclear masks/fluorescent images shown in Figure 1B-E for the four cell lines highlight the potential for variability in nuclear content/configuration. The calculation of nuclear and cytoplasmic area, and by extension the N:C ratio is based on the “diameter” feature of the image masks for the nucleus and cell – calculations with “diameter” is only mathematically consistent if the objects have high circularity. Can the authors clarify the circularity feature/masks in the IDEAS software?

Reviewer #2: Sebastian et al. analyze the nuclear-to-cytoplasmic ratios of four cell lines derived from malignancies using image-based flow cytometry to demonstrate the use of this technology in measuring N/C ratios. The technology appears to effectively calculate N/C ratio histograms for a large number of events, which would not be feasible using traditional light microscopy. The average N/C ratios varied between the malignant cell lines, as expected, but were typically above 0.6 which is consistent with what has been found for high grade urothelial carcinoma. The study is concisely written and accessible and covers a topic that is of great interest in the field of cytopathology, both academically and commercially. Despite its simplistic design and results that are not immediately actionable, the study has great novelty and is impactful to the field of cytomorphology (and, because of that, digital pathology).

There are some issues that the authors should address:

1. The authors largely misunderstand the cited references 9-11. While enlarged N/C ratio is one morphologic feature of malignancy, in general, that is seen in both histology and cytology, the N/C ratio is most easily determined in cytologic specimens and not histology, and is currently mostly used in the study of urinary cytology. While the authors correctly cite the references 9-11, they incorrectly say on page 3 that this is the examination of histologic sections. In fact, this is not true, as cytology specimens are not sectioned and are not referred to as "histology" - the slides are fixed to a glass slide using any number of methods (primarily alcohol-based fixation or air fixation). Therefore, while the entire 3D structure of the cell can be examined by adjusting the microscope focus, the N/C ratio is typically calculated in one “Z-plane” in which both the nuclear contour and cytoplasmic contour appear most in focus. The authors should adjust their text to take this into account, and it would be relevant to comment on how the flow cytometric method determines which “Z-plane” contains the calculated N and C diameter values.

2. The authors use only malignant cell lines and it would have been nice to have used non-malignant control cells as well, even if not clonally expanded.

3. The authors should note that most epithelial malignancies taken directly from a patient will have cohesive cells and tissue fragments. To use this method would require breaking the cells apart, which may alter the cellular natural state, or would be biased towards individual cells, which typically are more pleomorphic than cells that remain in fragments. However, the method is immediately useful in malignancies which tend to be single cells (e.g. urothelial carcinoma and lymphoma/leukemia).

4. The authors should mention that N/C ratio is an important cytomorphologic feature of malignancy for many, but not all, tissue types. For instance, melanoma tends to have more cytoplasm and lower N/C ratios than urothelial carcinoma. Carcinomas can have greatly varied N/C ratios. However, it is true that a very high N/C ratio (>0.7) is usually not seen in benign cells.

6. PLOS authors have the option to publish the peer review history of their article (what does this mean?). If published, this will include your full peer review and any attached files.

Reviewer #1: No

Reviewer #2: No

---

## [Author Response · Author response to Decision Letter 0]

20 Feb 2021

The response to Reviewer and Editor comments can be found in the attached file titled, "Sebastian_et_al_Rebuttals_SubmissionCopy.docx".

---

## [Decision Letter · Decision Letter 1]

16 Apr 2021

PONE-D-20-26034R1

An image-based flow cytometric approach to the assessment of the nucleus-to-cytoplasm ratio

PLOS ONE

Dear Dr. Sebastian,

Thank you for submitting your manuscript to PLOS ONE. After careful consideration, we feel that it has merit but does not fully meet PLOS ONE’s publication criteria as it currently stands. Therefore, we invite you to submit a revised version of the manuscript that addresses the points raised during the review process.

Due to the inavailability of one of the previous reviewers, we sent out your manuscript for a further evaluation by an additional independent reviewer which suggested to perform minor revision of the work.

Once the suggested modifications are addressed we will be glad to reconsider the paper for publication in the journal.

We look forward to receiving your revised manuscript.

Kind regards,

Vincenzo L'Imperio

Academic Editor

PLOS ONE

Journal Requirements:

Reviewers' comments:

Reviewer's Responses to Questions

**Review Comments to the Author**

Reviewer #2: I think the authors have satisfactorily answered the criticisms of the reviewers. Given their findings compared to now included normal cell control group, I think it is remarkable how closely the N/C cut-offs correlate with the findings in two studies of urothelial carcinoma in which image-based morphometric analysis was used. While I do not think the authors necessarily need to further modify their manuscript, the findings do, in general, support the authors' findings. And, if the authors wish to pursue this area further, discussion in these manuscripts may provide valuable:

1. Digital image analysis supports a nuclear-to-cytoplasmic ratio cutoff value of 0.5 for atypical urothelial cells. PMID: 28581671 DOI: 10.1002/cncy.21883

2. Digital image analysis supports a nuclear-to-cytoplasmic ratio cutoff value below 0.7 for positive for high-grade urothelial carcinoma and suspicious for high-grade urothelial carcinoma in urine cytology specimens. PMID: 30395388 DOI: 10.1002/cncy.22061

Reviewer #3: The Authors describe the application of IFC to the measurement of the N:C ratio in one benign and four malignant human cell lines.

The methods and their description, as well as the overall manuscript style and language, are irreproachable.

However, my impression is that the study suffers from the absence of a pathologist/physician among the Authors (both in the overall design and in the manuscript itself).

Let us consider the main ideas of the study separately:

1. IFC has potential as a technique to measure N:C

The Authors convincingly show that IFC can be used to measure the N:C ratio of a population of cells. This idea is original and supported by the study design and data.

However, the practical applications of this are arguable because histology (especially if empowered by computer-aided image analysis) might be better, considering that it is already routinely performed on all human cancers and does not require other tools (like an IFC).

2. the N:C ratio (measured by IFC) can be used as a discriminator of malignancy

In my opinion, this premise is flawed. It will be discussed below.

--------

56: "Clinicians have identified an enlarged nucleus as one of the most prevalent characteristics of malignant cells"

An enlarged nucleus is definitely a characteristic of some neoplasms. However, it is neither sensitive nor specific. There are benign cells with huge nuclei (think symplastic leiomyoma) and malignant cells with tiny nuclei (signet ring cell cancers, foamy prostate cancer). Similarly, the N:C ratio is increased in most cancers, but again it is neither sensitive nor specific. There are benign cells with alarmingly high N:C ratios (lymphocytes) and malignant cells with very low N:C ratios (again, signet ring cells). Furthermore, some neoplasms are by definition cytologically indistinguishable from their benign counterparts (and thus they share an identical N:C ratio). This is the case of follicular thyroid adenoma/carcinoma and several other endocrine neoplasms.

211: "The N:C ratio is widely used as a histological metric in staging malignant disease in most tissue types". This is not true, as far as I'm aware. Histological grading (not staging) of cancers depends on the histotype, but only rarely does the N:C ratio make an appearance in grading criteria. For example, prostate cancer is graded based on architecture alone; endometrial, colorectal, and lung cancers are graded mostly based on architecture; even in neoplasms in which the N:C ratio is somehow considered in grading (urothelium, breast), it is just one of several criteria.

212: "An enlarged nucleus has become a hallmark in tumor staging and grading due to an abundance of chromatin present within malignant cells". Staging and grading are two distinct things. Grading has been discussed previously. Staging of most if not all human cancers uses the AJCC TNM system which does not take the N:C ratio into account at all.

Minor points:

- 48: the Authors seem to suggest that histology as the gold standard technique to diagnose malignancy should be replaced by other, high-throughput, techniques. This is arguable, but not supported by the rest of the paper.

- 240: what's TPM?

- 250: histological sectioning is indeed tedious, but it is performed by routine on all human cancers, so the added cost would be minimal.

- Is the N:C ratio calculated as a ratio of diameters, areas or volumes?

- 59: Measuring the N:C ratio by visual approximation on histological images is indeed error-prone and operator-dependent. However, computer-aided image analysis can overcome this problem.

- 214: "Currently, histology is the gold standard assessment method for the determination of the N:C ratio but it cannot be practically used when analyzing large populations of cells". Partly true: digital pathology and computer-aided image analysis are overcoming this problem. The Authors themselves disprove this statement at line 225.

In conclusion, IFC is an interesting and promising technique, and it can surely be used to measure the N:C ratio of cells (as this manuscript shows). However, the idea of diagnosing malignancy based on the N:C ratio (measured by IFC or otherwise) seems flawed. The study would be better reorganized by focusing on the N:C measurement by IFC as the main finding and discarding any implications on the diagnosis of malignancy.

---

## [Author Response · Author response to Decision Letter 1]

11 May 2021

All Review rebuttals can be found in the document "Sebastian_et_al_Rebuttals_2_SubmissionCopy.docx".

---

## [Decision Letter · Decision Letter 2]

26 May 2021

PONE-D-20-26034R2

An image-based flow cytometric approach to the assessment of the nucleus-to-cytoplasm ratio

PLOS ONE

Dear Dr. Sebastian,

Thank you for submitting your manuscript to PLOS ONE. After careful consideration, we feel that it has merit but does not fully meet PLOS ONE’s publication criteria as it currently stands. Therefore, we invite you to submit a revised version of the manuscript that addresses the points raised during the review process.

After a re-evaluation of the revised manuscript, one of the reviewer raised additional observations that could be addressed to further improve the quality of the paper. We'll be glad to receive the new version of the work after the corrections suggested for the final decision on approval.

We look forward to receiving your revised manuscript.

Kind regards,

Vincenzo L'Imperio

Academic Editor

PLOS ONE

Journal Requirements:

Reviewers' comments:

Reviewer #2: The authors have appropriately addressed all comments. However, I want to make a few corrections in their response and modifications.

- For the new statement “Our N:C results are consistent with international standards of cytopathology (such as Zhang et al.’s (10) finding of an N:C ratio ≥0.7 as malignant, Hang et al.’s (42) finding of an N:C ratio cutoff value of 0.5 for atypical urothelial cells, and McIntire et al.’s (43) finding of a N:C ratio cutoff value below 0.7 for high-grade urothelial carcinoma).” please remove reference 10 here. Reference 10 is correctly cited earlier in the manuscript as it studied interobserver variability, but it did not study the impact of N/C ratio on determining whether cells were malignant or not.

- Regarding the comments of reviewer #3, I agree that the N/C ratio is not in general a standard used in histopathologic examination for grading malignancies. It is true that it plays a role in some types of malignancy as increased N/C ratio is one of many features used to assess a cell for cytologic atypia. I bring this up because the authors seem to focus mostly on N/C ratio in tissue sections when the assessment is more readily done in cytologic specimens, and in cytology the only formal system that requires an N/C ratio assessment is The Paris System for urine specimens. So, while the N/C ratio is one of many cytologic features that can help diagnose malignancy or grade/differentiation of a malignancy, it depends on the specimen type and the cancer type. I would recommend double checking the manuscript to ensure the introduction and discussion is consistent with this idea. In part, because the N/C ratio can be difficult to assess in both histology and cytology, is is important that the authors have shown a way to objectively measure the N/C ratio and correlate ratios with malignant vs. benign cells. This can help show the degree to which N/C ratio variations are associated with malignancy, at least in certain cell types, even if no single feature can be used, in most cases, to determine whether a cell is malignant or benign.

Reviewer #3: All comments have been addressed in a satisfactory manner.

State-of-the-art high-throughput whole slide scanners coupled with appropriate software are already usable to get N:C data in high-throughput contexts; however I have to agree that until these become ordinary, they represent an extra cost (not unlike an IFC).

---

## [Author Response · Author response to Decision Letter 2]

26 May 2021

The Reviewer replies is in a document titled, "Sebastian_et_al_Rebuttals_3_SubmissionCopy".

---

## [Decision Letter · Decision Letter 3]

7 Jun 2021

An image-based flow cytometric approach to the assessment of the nucleus-to-cytoplasm ratio

PONE-D-20-26034R3

Dear Dr. Sebastian,

We’re pleased to inform you that your manuscript has been judged scientifically suitable for publication and will be formally accepted for publication once it meets all outstanding technical requirements.

Kind regards,

Vincenzo L'Imperio

Academic Editor

PLOS ONE

Reviewers' comments:

Reviewer #2: All comments have been addressed

---

## [Editor Report · Acceptance letter]

16 Jun 2021

PONE-D-20-26034R3 

An image-based flow cytometric approach to the assessment of the nucleus-to-cytoplasm ratio 

Dear Dr. Sebastian:

I'm pleased to inform you that your manuscript has been deemed suitable for publication in PLOS ONE. Congratulations! Your manuscript is now with our production department. 

Kind regards, 

on behalf of

Dr. Vincenzo L'Imperio 

Academic Editor

PLOS ONE